# Relative Importance of Land Use and Climate Change on Hydrology in Agricultural Watershed of Southern China

**Lanhua Luo [1,2], Qing Zhou [1], Hong S. He [3], Liangxia Duan [1], Gaoling Zhang [1] and Hongxia Xie [1,***

[1] College of Resources and Environment, Hunan Agricultural University, Changsha 410128, China; 181302085@stu.njnu.edu.cn (L.L.); zhouqing@hunau.edu.cn (Q.Z.); duanliangxia@hunau.edu.cn (L.D.); zhanggaoling@stu.hunau.edu.cn (G.Z.)

[2] Key Laboratory of Virtual Geographic Environment, Nanjing Normal University, Ministry of Education, Nanjing 210023, China

[3] School of Natural Resources, University of Missouri-Columbia, Columbia, MO 65211, USA; heh@missouri.edu

* Correspondence: hxxie@hunau.edu.cn; Tel.: +86-1597-319-6669; Fax: +86-0730-8461-7803

**Abstract:** Quantitative assessment of the impact of land use and climate change on hydrological processes is of great importance to water resources planning and management. The main objective of this study was to quantitatively assess the response of runoff to land use and climate change in the Zhengshui River Basin of Southern China, a heavily used agricultural basin. The Soil and Water Assessment Tool (SWAT) was used to simulate the river runoff for the Zhengshui River Basin. Specifically, a soil database was constructed based on field work and laboratory experiments as input data for the SWAT model. Following SWAT calibration, simulated results were compared with observed runoff data for the period 2006 to 2013. The Nash-Sutcliffe Efficiency Coefficient (NSE) and the correlation coefficient ($R^2$) for the comparisons were greater than 0.80, indicating close agreement. The calibrated models were applied to simulate monthly runoff in 1990 and 2010 for four scenarios with different land use and climate conditions. Climate change played a dominant role affecting runoff of this basin, with climate change decreasing simulated runoff by −100.22% in 2010 compared to that of 1990, land use change increasing runoff in this basin by 0.20% and the combination of climate change and land use change decreasing runoff by 60.8m³/s. The decrease of forestland area and the corresponding increase of developed land and cultivated land area led to the small increase in runoff associated with land use change. The influence of precipitation on runoff was greater than temperature. The soil database used to model runoff with the SWAT model for the basin was constructed using a combination of field investigation and laboratory experiments, and simulations of runoff based on that new soil database more closely matched observations of runoff than simulations based on the generic Harmonized World Soil Database (HWSD). This study may provide an important reference to guide management decisions for this and similar watersheds.

**Keywords:** land use change; runoff; climate change; SWAT model

## 1. Introduction

River runoff is an important fresh water resource that directly affects water resource utilization and maintenance. The Fifth Assessment Report of the Intergovernmental Panel on Climate Change (IPCC) pointed out that surface runoff in high latitudes and tropical humid regions will increase in the 21st century, whereas surface runoff will decline in most subtropical arid regions and in Mediterranean regions [1]. In the past few decades, the runoff of most rivers in China has declined [2,3]. Changes in

land use and climate are primary factors influencing changes in runoff [4–6]. Quantifying the impact of land use and climate changes on runoff provides essential guidance for developing new policies and taking management action.

Land use change including deforestation, afforestation, and the development of agriculture and urban areas are common human activities that have important effects on hydrological processes [7,8]. Land use change influences land cover types, changes the production of surface runoff, and thus affects the hydrological processes of the basin where the changes occur. Generally, forests conserve water sources and reduce surface runoff, while deforestation can increase surface runoff [9–11]. Urbanization or other human development often increases impervious surfaces and causes increased surface runoff [12–14].

The influence of climate change on runoff mainly depends on the associated changes in precipitation and evaporation [15,16]. Chang [17] found that runoff showed great sensitivity to precipitation change in the Conestoga River Basin of Pennsylvania. Muyibul et al. [18] found that in the upper Urumqi River watershed the influence of climate change on runoff was greater than that of land use change; runoff had a positive correlation with precipitation change and a negative correlation with temperature change. Fontaine et al. [19] simulated hydrological response to climate change in the Montenegro region of South Dakota, USA. Under different climate scenarios, they found when the temperature increased by 4 °C that the annual runoff decreased by 39%, and when rainfall increased by 10%, the annual runoff increased by 44%. These studies suggested that runoff increases with increasing precipitation and decreases with increasing temperature.

The hydrological response to land use and climate changes is complex and the previous studies have also shown that there are significant differences in the degree of impact of land use and climate changes on runoff under different river characteristics [15]. Distinguishing the influence of the two factors and quantitatively analyzing their respective effects on hydrological processes are challenging. In addition, studies have shown that hydrological responses to climate and land use changes may differ even for different tributaries of the same river [20]. Dai et al. [21] stated that the relative importance of land use change to runoff in the middle reaches of the Yangtze River was greater than that of climate change. Liu et al. [22] showed that precipitation was the main factor influencing runoff of the Ganjiang River, one of the main tributaries of the middle and lower reaches of the Yangtze River, but the effect on runoff was not obvious for land use change due to human activities.

The SWAT (Soil and Water Assessment Tool) model is the most commonly used hydrological assessment model [23–31]. Chaplot [32] applied SWAT to analyze effects of climate change on water and soil resources in two agriculture watersheds. Ghaffari et al. [33] used SWAT to investigate the hydrological effects of land use change in the Zanjanrood basin. In China, Lin et al. [34] used SWAT to analyze the impacts of land use change on runoff in the JinJiang catchment. Li et al. [35] applied SWAT to quantify changes in annual runoff due to precipitation variability, land use change and construction of check dams. Luan et al. [36] applied SWAT to assess the effects of land use changes on major hydrologic processes for a large irrigation district within a plains landscape. These studies suggest that the SWAT model has been effectively adapted in many watersheds of China.

Certain soil parameters are required for application of the SWAT model, and the precision of the soil parameters affects the accuracy of the simulations [37]. Lack of suitable soil parameter estimates is a major factor restricting widespread SWAT applications. By default, initial values of the soil parameters for the SWAT model are calculated based on monitoring data from Temple Station in Texas, USA, which is quite different from the actual situation in other research areas. Therefore, in each new study area, it is especially important to establish a soil database that is in line with the actual situation. For applications in China, some scientists have used the China soil database of the Nanjing Institute of Soil Science of the Chinese Academy of Sciences to establish soil databases for use with the SWAT model [38]. The soil particle size standard for the SWAT model is the American Standard [39]. However, the Chinese soil database of the Nanjing Institute of Soil Sciences of Chinese Academy of Sciences uses the International system [38]. Thus, it is necessary to convert the international standard of soil

particle size into the American standard. Some scientists have also established soil databases using the HWSD (Harmonized World Soil Database) published by the Food and Agriculture Organization of the United Nations (FAO) [40,41]. This method can avoid errors caused by the soil particle size conversion, but the resolution of the soil data is 1:1,000,000, which affects simulation accuracy in small watersheds. Prior research results showed that soil data with higher resolution had relatively good simulation results [42]. Therefore, it may be necessary to construct more detailed soil databases based on field measurements to get a good simulation result.

Previous studies have shown the applicability of the SWAT model in the Xiangjiang River Basin and that increasing forest and grassland area would reduce runoff, while increasing the area of cultivated land and developed land would increase runoff [43]. The Zhengshui River Basin, the primary tributary of the Xiangjiang River, has some serious soil erosion problems. Quantitatively analyzing causes of runoff change in the Zhengshui River Basin is critical for managing land use, water quality and water use in this region. Du et al. [44] used a mathematical statistics method to analyze the impact of precipitation change and human activities on the hydrology of the Zhengshui River Basin. In a prior study of the Zhengshui River Basin, He [45] studied the effect of reservoir operation on hydrological processes by using the SWAT model, the result showed that the SWAT model was suitable for application to the Zhengshui River Basin. In He's study [45], the soil texture data of the soil database were acquired from volume three of "Chinese Soil Genus Records" and required soil particle size conversions. More detailed, site-specific estimates of soil parameters would likely result in more accurate simulation of runoff in the Zhengshui River Basin.

Thus, the objectives of this study of the Zhengshui River Basin were as follows: (1) calibrate the SWAT model for the Zhengshui River Basin; (2) compare the accuracy of the simulation results based on a new, detailed soil database constructed by field investigation and laboratory experiments with results based on the relatively coarse HWSD; (3) quantify the impact of land use and climate changes on runoff by applying the calibrated SWAT model under different land use and climate scenarios.

## 2. Materials and Methods

### 2.1. Study Area

The Zhengshui River Basin covers an area of 3470 km$^2$ and is located in Hunan province with longitude ranging from 111°53′ E to 112°37′ E and latitude ranging from 26°52′ N to 27°10′ N. The Zhengshui River is a first-order tributary of the Xiangjiang River and has a river length of 194 km. The river originates in the southeast of Shaodong county, flows through Shaoyang City, Loudi City and Hengyang City, and then converges into the Xiangjiang River at Hengyang City. The upper reaches of the basin are surrounded by mountains with a land area of 2720 km$^2$ and accounting for about 71% of the total area of the basin. The middle and lower reaches of the basin are dominated by plains and lowlands with an area of 750 km$^2$, accounting for about 20% of the total basin area. The remainder is the water area. The Zhengshui River Basin has a subtropical monsoon climate. The average temperature in July is above 29 °C. The annual precipitation is about 1300 mm, and is concentrated in the spring and summer, often arriving in the form of storms. The parent material of the soil in the river basin is sand-shale and granite, which is severely weathered. Soil erosion is severe and often creates blockages to river flow. The water resources in Zhengshui River Basin are currently at medium to high levels of utilization and cannot sustain further large-scale development [46]. There are seven weather stations within or close to the basin and two hydrological stations (Shenshantou station and Shimenkan station) (Figure 1). As the observed runoff data of the Shimenkan station are not complete, this study only used the Shenshantou station data for calibration and validation of the SWAT model.

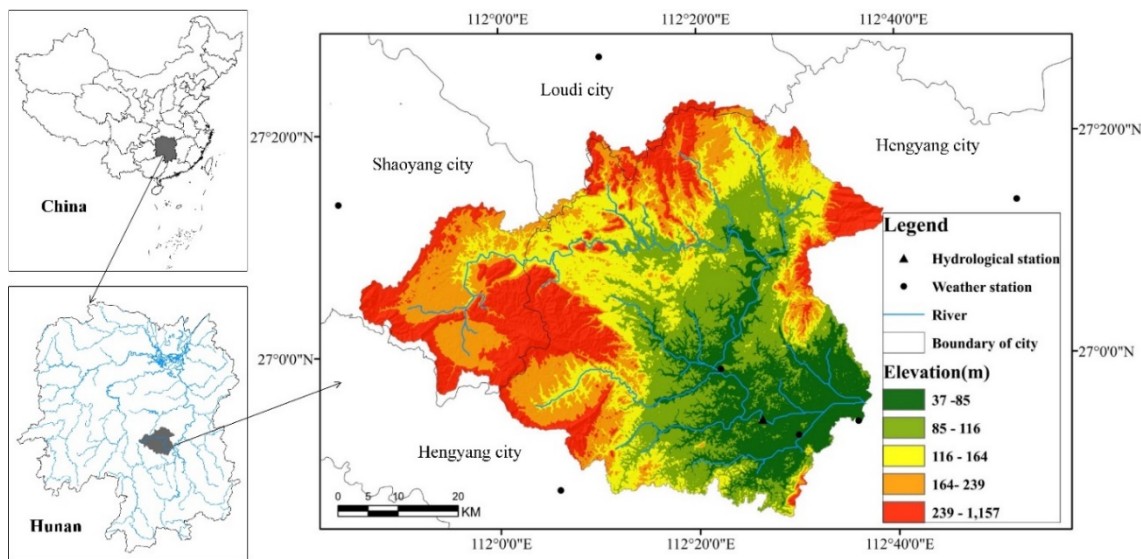

**Figure 1.** Location of the Zhengshui River Basin showing hydrological and weather stations.

## 2.2. Data

SWAT model simulation results are very sensitive to the input data. In this study, the data used in the SWAT model included a 17-m-resolution digital elevation model (DEM), a 30-m-resoluton land use map, a 1:50,000 soil distribution map with soil physical-chemical properties from the Hunan Provincial Soil Survey Project, weather data from surrounding weather stations, and monthly discharge data from the Shenshantou Hydrological Station (Table 1).

**Table 1.** Data used in the study.

| Datasets | Resolution | Format | Source |
|---|---|---|---|
| Digital elevation model (DEM) | $17 \times 17$ m | Grid | Google Earth |
| Land use data | $30 \times 30$ m | Grid | Data Center for Resources and Environmental Sciences of Chinese Academy of Sciences, Institute of geographic sciences and natural resources research |
| Soil map | 1:50,000 | Shape file | Hunan Agricultural University |
| Soil attributes of different soil profile | point | xls | Hunan Provincial Soil Survey Project, Hunan Agricultural University |
| Weather data | Daily | xls | Hunan Provincial Key Laboratory of Meteorological Disaster Prevention and Reduction |
| Discharge data | Monthly | xls | Hunan Provincial Water Resources Department |

### 2.2.1. Soil Data

Soil data included a soil type map and physical-chemical properties of the associated soil. The scale of soil type map used in this study was 1:50,000. The classification of soil types of the soil type map is based on Soil Taxonomy in China. There are six levels of Chinese Soil Taxonomy. Considering the unity with the data from soil profile that are necessary parameters for runoff simulation, we adopt the classification of subgroup. The soil in the Zhengshui River Basin was divided into 15 types. The most widespread soil in the Zhengshui River Basin was red soil, with an area of 1423 km$^2$, accounting for 41% of the total area of the basin, followed by waterlogged paddy soil, also accounting for nearly 41% of the total area of the basin. The area of mountain shrubby-meadow soil was the smallest, only 0.06 km$^2$, accounting for a small proportion of the total area of the basin. 199 soil profiles were excavated in order to classify the soil system of Hunan Province, in the Hunan Provincial Soil Survey Project. Parameters of soil profile for this study were obtained from the Hunan Provincial Soil Survey

Project. Through field observation and analysis, the number of soil layers in each soil type (NLAYERS), the depth from the bottom layer to the surface layer of soil layer (SOL_Z), the maximum root depth of soil profile (SOL_ZMX), and the structure of soil layer (TEXTURE) were recorded. Each soil layer in each soil profile was sampled and measured in the laboratory to obtain the chemical parameters of soil. The parameters of organic carbon content (SOL_CBN), clay content, silt content, sand content and rock content required by the database were obtained by experiments, especially the particle size, which was in accordance with the American Standard, which meets the needs of the model, and did not need soil particle size conversion. The USLE soil erodibility factor (K) parameter was calculated by using EPIC model. Groupie of pedohydrology (HYDGRP) was divided according to the size of soil permeability coefficient, and soil permeability coefficient was calculated according to empirical calculation formula of soil permeability coefficient. The parameters related to soil moisture including wet bulk density (SOL_BD), effective water content (SOL_AWC) and saturated hydraulic conductivity (SOL-K) were calculated with the soil-water-characteristics module (SWCT) in soil-plant-atmosphere-water model (SPAW) [47]. The parameters of anion exchange porosity (ANION_EXCL), maximum compressibility of soil (SOL_CRK), surface reflectance (SOL_ALB) and soil electrical conductivity (SOL_EC) were the default values of the model. With the aforesaid data, the soil database of the Zhengshui River Basin was constructed.

In order to evaluate the impact of different soil databases on the simulation results of the SWAT model in the Zhengshui River Basin. Another soil database based on the HWSD database of the Zhengshui River Basin was constructed (Figure 2). To evaluate the differences in SWAT simulations, we used these two soil databases as input data and the same climate and land use data, and simulated the monthly runoff of the Zhengshui River Basin from 1999 to 2005. We then calibrated the SWAT model from the soil data before making predictions.

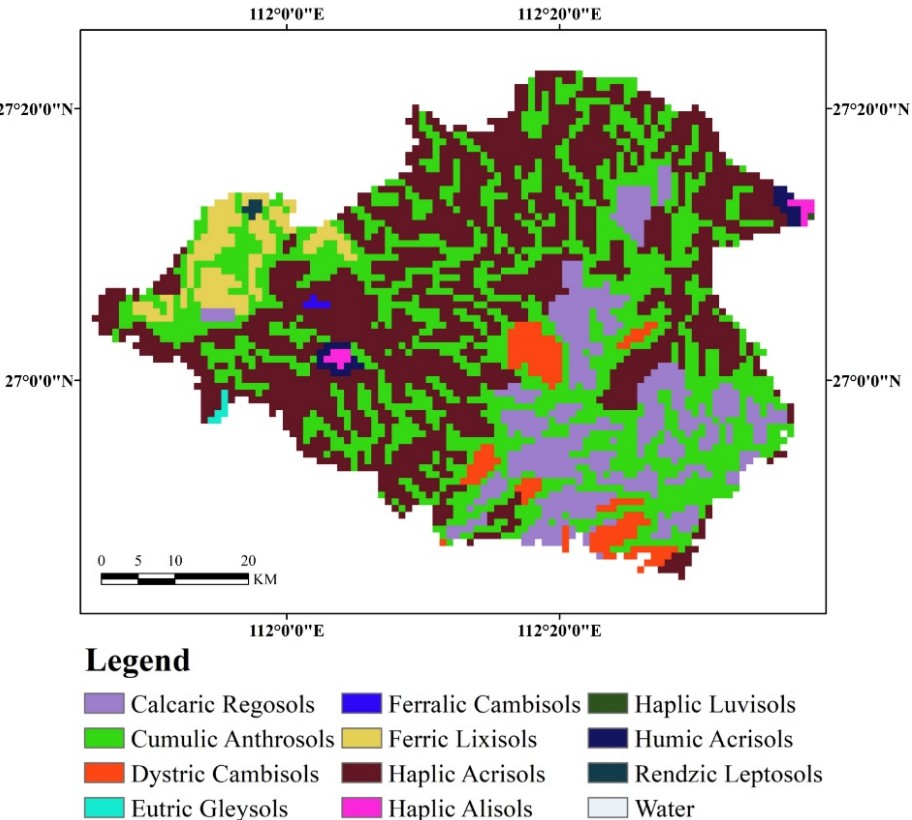

**Figure 2.** The map of soil types of the Zhengshui River Basin based on HWSD (Harmonized World Soil Database).

### 2.2.2. Land Use Data

The land use data with 30 m spatial resolution were obtained from the Data Center for Resources and Environmental Sciences of the Chinese Academy of Sciences (RESDC). The land use types in the Zhengshui River Basin included cultivated land, forestland, grassland, water bodies, developed land, and unutilized land. From 1990 to 2010, the area of cultivated land and forestland accounted for a large proportion of the total area of Zhengshui River Basin, up to 90%. The area of unutilized land was the smallest (Table 2). The area of cultivated land and forestland decreased over the 20-year observation period, while the area of grassland, water bodies and developed land increased and the area of unutilized land was essentially unchanged. The area of developed land increased 13.0 km$^2$ (27.9%). The area of cultivated land decreased 13.0 km$^2$. Despite the large area of cultivated land change, the proportion of change was only 0.9%. The percent change of grassland and water body was 10.4% and 8.4%, respectively. However, the area of these two land use types was still small. To further understand the impact of land use on runoff, the area changes of cultivated land, developed land and forestland in each sub-basin from 1990 to 2010, and the corresponding runoff change were analyzed.

**Table 2.** Land use and its change in the Zhengshui River Basin from 1990 to 2010.

|  | Cultivated Land | | Forestland | | Grassland | | Water Bodies | | Developed Land | | Unutilized Land | |
|---|---|---|---|---|---|---|---|---|---|---|---|---|
|  | Area (km$^2$) | % | Area (km$^2$) | % | Area (km$^2$) | % | Area (km$^2$) | % | Area (km$^2$) | % | Area (km$^2$) | % |
| 1990 | 1495.8 | 43.0 | 1882.3 | 54.1 | 19.2 | 0.6 | 37.4 | 1.1 | 46.7 | 1.3 | 0.1 | 0.0 |
| 2010 | 1482.8 | 42.6 | 1881.0 | 54.0 | 17.2 | 0.5 | 40.6 | 1.2 | 59.7 | 1.7 | 0.1 | 0.0 |
| Change | −13.0 | −0.9 | −1.2 | −0.1 | −2.0 | −10.4 | 3.1 | 8.4 | 13.0 | 27.9 | 0.0 | 0.0 |

### 2.2.3. Weather Data

The weather data used in the SWAT model were obtained from seven weather stations located in or near the study area from 1961 to 2014. The weather data required by the SWAT model included daily precipitation, daily maximum and minimum temperatures, wind speed, relative humidity, evapotranspiration and solar radiation.

To quantitatively analyze runoff response to climate change, runoff was simulated by SWAT under different climate scenarios in 1990 and 2010. The monthly precipitation in 2010 (105 mm) was less than that in 1990 (120 mm). Both the highest and lowest monthly temperatures in 2010 were higher than in 1990, with an average monthly temperature of 18.2 °C in 2010 compared to 17.8 °C in 1990.

### 2.2.4. Discharge Data

Monthly observed runoff data of river discharge were obtained from the Shenshantou Hydrological Station of the Zhengshui River Basin. The observed runoff values were used for calibration and validation of the SWAT model. In this study, the model was calibrated with discharge data from 1999 to 2005 and validated with data from 2006 to 2013.

### 2.3. SWAT Model Description and Parameterization

The SWAT model is a continuous, long-term, physically-based, spatially-distributed model developed to predict the effects of climate and land management on the water cycle, sediment, and agricultural pollution transport in complex watersheds with different soil, land use, and management conditions [48–50]. The model divides a basin into sub-basins which can be further delineated into Hydrologic Response Units (HRUs). Each HRU is a combination of land use, soil and slope type. Flows generated from each HRU in a sub-basin are then summed and routed through channels. The SWAT model can simulate precipitation, infiltration, surface runoff, evapotranspiration, lateral flow, and percolation. Surface runoff is calculated using a modification



of the U.S. Soil Conservation Service (SCS, now called the Natural Resources Conservation Service, NRCS) curve number method with daily rainfall amounts [51]. In this study, the SWAT model for the study area was constructed using ArcSWAT [39].

The basin and sub-basin boundaries, as well as stream networks were delineated using DEM data. HRUs are the minimum unit of the model. When the model is operating, the hydrological processes of each HRU in the sub-basin are calculated, and then the output of all HRUs is combined at the outlet of the sub-basin to obtain the composite sub-basin output [52]. Soil properties and weather data were organized into a database for model input according to the requirements of the SWAT model. In this study, the basin was divided into 53 sub-basins based on the threshold area of 3,900 ha and the basin outlet added manually, which could result a drainage network closest to the actual situation. These sub-basins were further divided into 671 HRUs defined by similarity in their land use, soil subgroup type and slope.

## 2.4. Parameter Calibration and Validation

Sensitivity analysis of the model parameters can identify the parameters that have a great influence on the results of the model simulation. Attention to calibration of those parameters can reduce model uncertainty and improve the reliability of the model simulations. In this study, an automatic parameter estimation procedure, SWAT-CUP (SWAT-Calibration and Uncertainty Procedures) was used to analyze parameter sensitivity and calibrate the model parameters [53]. Five algorithms in SWAT-CUP and SUFI-2 (Sequential Uncertainty Fitting Version 2) were used in this SWAT calibration analysis. SUFI-2 is a tool for sensitivity analysis, multi-point calibration and uncertainty analysis. It can analyze multiple parameters. Compared to four alternative algorithms, SUFI-2 requires fewer model runs to achieve similarly good calibration and prediction uncertainty results [54]. After multiple iterations, the optimal parameters sets were determined. The calibrated parameters were selected referring to previous work [43,55]. In this study, 11 parameters were calibrated: SCS runoff curve number factor (CN2), available water capacity of the soil layer (SOL_AWC), groundwater hysteresis coefficient (GW_DELAY), soil layer depth (SOL_Z), shallow groundwater depth threshold value (GWQMN), base flow alpha factor (ALPHA_BF), groundwater "revap" coefficient (GW_REVAP), threshold of evaporation in shallow aquifer (PEVAPMN), saturated hydraulic conductivity (SOL_K), mean gradient (HRU_SLP), and soil evaporation compensation factor (ESCO).

A global sensitivity analysis was implemented for the selected model parameters. T-stat and P-Value were used to evaluate the significance of the relative sensitivity of parameters. The former index measures the sensitivity of a parameter; the larger the absolute value is, the more sensitive the parameter will be. The latter index determines the significance of the sensitivity; if it is closer to 0, the parameter will be more sensitive. After the calibration was finished, the parameters were adjusted manually for runoff simulation in the model, based on the optimal parameter set of the SWAT model. Observed runoff and simulated runoff were compared to evaluate model performance.

An array of statistical techniques can be used to evaluate SWAT hydrologic predictions. For example, Coffey et al. [56] described nearly 20 potential statistical tests that can be used to judge SWAT predictions performance, including coefficient of determination ($R^2$), Nash-Sutcliffe efficiency (NSE), root mean square error (RMSE), nonparametric tests, *t*-test, objective functions, autocorrelation and cross-correlation. Among them, $R^2$ and NSE are widely used. They indicate the similarity between the simulated runoff and observed runoff. Therefore, $R^2$ and NSE were used to evaluate the model performance in this study. NSE and $R^2$ are calculated as follows:

$$R^2 = \frac{\left[\sum_{i=1}^{n}\left(Q_m - \overline{Q_m}\right)\left(Q_s - \overline{Q_s}\right)\right]^2}{\sum_{i=1}^{n}\left(Q_m - \overline{Q_m}\right)^2 \sum_{i=1}^{n}\left(Q_s - \overline{Q_s}\right)^2} \tag{1}$$

$$\text{NSE} = 1 - \frac{\sum_{i=1}^{n}(Q_m - Q_s)^2}{\sum_{i=1}^{n}(Q_m - \overline{Q_m})^2} \tag{2}$$

where $Q_m$ and $Q_s$ are the measured and simulated monthly discharge values, respectively; $\overline{Q_m}$ and $\overline{Q_s}$ are the average measured and simulated discharge values, respectively; and $n$ is the number of simulated and observed data pairs. NSE ranges from $-\infty$ to 1 and $R^2$ from 0 to 1. The closer NSE and $R^2$ are to 1, the better the performance of a model in simulating observed data. To date, no absolute criteria have been firmly established in the literature for judging adequate model performance based on $R^2$ and NSE [57]. Liu et al. [58] proposed that when NSE is less than 0, the model results are unacceptable. The simulation results are generally considered acceptable when NSE is greater than 0.54 and model performance is considered very good if NSE is greater than 0.65.

### 2.5. Differentiation of Effects of Land Use and Climate Changes on Runoff

To quantitatively analyze the response of runoff to land use change and climate change, four scenarios ($S_1$ to $S_4$) were analyzed for change in runoff. $S_1$ represented climate in 1990 ($C_{1990}$) with land use in 1990 ($L_{1990}$); $S_2$ represented climate in 2010 ($C_{2010}$) with land use in 2010 ($L_{2010}$); $S_3$ represented climate in 2010 ($C_{2010}$) with land use in 1990 ($L_{1990}$); and $S_4$ represented climate in 1990 ($C_{1990}$) with land use in 2010 ($L_{2010}$). $R_{S1}$, $R_{S2}$, $R_{S3}$ and $R_{S4}$ indicated the simulated runoff under scenarios $S_1$, $S_2$, $S_3$ and $S_4$ respectively. Change in the amount of runoff caused by land use and climate change together was calculated as:

$\Delta R_{L\&C} = R_{S2} - R_{S1}$, where $\Delta R_{L\&C}$ indicates runoff change caused by land use and climate change. Furthermore:

$\Delta R_C = R_{S3} - R_{S1}$, $\Delta R_C$ represents runoff change caused by climate change;

$\Delta R_L = R_{S4} - R_{S1}$, $\Delta R_L$ represents runoff change caused by land use change.

The relative (percent) contribution of land use and climate change to total runoff change was calculated as follows:

$I_L = \Delta R_L / \Delta R_{C\&L}$ *100%, $I_L$ estimates the land use change effect on runoff;

$I_C = \Delta R_C / \Delta R_{C\&L}$ *100%, $I_C$ estimates the climate change effect on runoff.

## 3. Results

### 3.1. Model Sensitivity, Calibration, and Validation

Runoff estimates from the SWAT model were calibrated and validated at the month scale. In this study, model sensitivity was evaluated for 11 different parameters (Table 3). According to the SWAT T-stat and P-Value, parameters were ranked by sensitivity from 1 (most sensitive) to 11 (least sensitive). The sensitivity analysis results indicated that for the Zhengshui River Basin, the parameters for SOL_K, CN2, and SOL_AWC were more sensitive than other parameters.

The NSE and $R^2$ for the calibration period (1990 to 2005) were 0.83 and 0.83, respectively, and NSE and $R^2$ were 0.84 and 0.83 for the validation period (2006–2013). Values of NSE and $R^2$ were greater than 0.8, indicating satisfactory performance of the calibrated SWAT model. The comparison between simulated and observed monthly runoff values in the periods of calibration and validation confirmed that the simulated and observed values were closely aligned and that the model effectively simulated monthly runoff in the Zhengshui River Basin (Figure 3a,b).

For the non-calibrated model, the simulated results showed that the monthly runoff based on the soil database derived from the HWSD database was higher than that based on the local soil dataset. The runoff predicted by both of the soil datasets was higher than the observed monthly runoff data (Figure 3c). NSE and $R^2$ were used to compare the goodness-of-fit between simulated and observed data. The NSE values obtained were 0.50 and 0.12 for the soil database constructed by field investigation and laboratory experiment, and the soil database constructed from the HWSD database, respectively. The $R^2$ values were 0.77 and 0.69, respectively (Figure 4a,b). Thus, the soil

database constructed by field investigation and laboratory experiment predicted runoff reasonably well before calibration.

**Table 3.** List of sensitive parameters selected for calibration and the ranges and values of parameters used.

| Parameter | Description | Initial Calibration Range | Sensitivity Ranking | Calibration Value |
|---|---|---|---|---|
| r_SOL_K.sol | Saturated hydraulic conductivity | (−1, 1) | 1 | 0.98 |
| r_CN2.mgt | SCS runoff curve number factor | (−1, 1) | 2 | −0.28 |
| r_SOL_AWC.sol | Available water capacity of the soil layer | (−1, 1) | 3 | 0.47 |
| v_HRU_SLP.hru | Mean gradient | (0, 1) | 4 | 0.96 |
| r_SOL_Z.sol | Soil layer depth | (−1, 1) | 5 | 0.67 |
| v_GW_REVAP.gw | Groundwater "revap" coefficient | (0.02, 0.2) | 6 | 0.91 |
| v_GWQMN.gw | Shallow groundwater depth threshold value | (0, 5000) | 7 | 3553.07 |
| v_ALPHA_BF.gw | Base flow alpha factor | (0, 1) | 8 | 0.37 |
| v_GW_DELAY.gw | Groundwater hysteresis coefficient | (0, 500) | 9 | 105.82 |
| v_ESCO.hru | Soil evaporation compensation factor | (0, 1) | 10 | 1 |
| v_REVAPMN.gw | Threshold of evaporation in shallow aquifer | (0, 500) | 11 | 193.43 |

v_ in the parameter name means that the default parameter is replaced by the calibration value or absolute change, r_ represents the default parameter value multiplied by (1 + calibration value) or relative change (i.e., CN2_new = CN2_old*(1 + CN2_calibration value)) [59].

SWAT model calibration generally improves the reliability of the model predictions [42]. SWAT-CUP was used to calibrate the corresponding SWAT output of the two soil databases and get the best simulation results. After calibration, the monthly runoff based on the local soil dataset is still closer to the observed runoff (Figure 3d). The NSE values obtained were 0.83 and 0.66 for the soil database constructed by field investigation and laboratory experiment, and the soil database constructed from the HWSD database, respectively. The $R^2$ values obtained were 0.83 and 0.76, respectively (Figure 4c,d). Both databases provided satisfactory estimates after SWAT model calibration, but the soil database constructed from field investigation and laboratory experiments produced a better estimates.

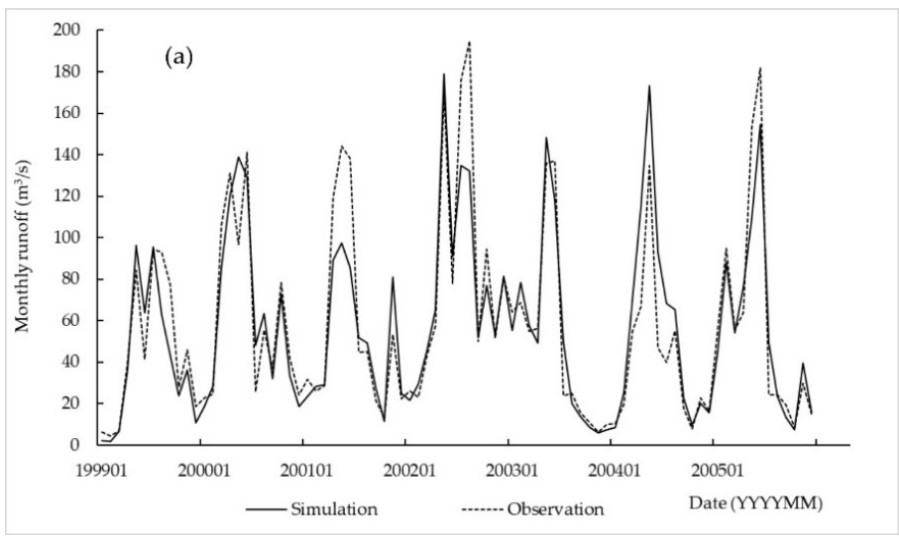

**Figure 3.** *Cont.*

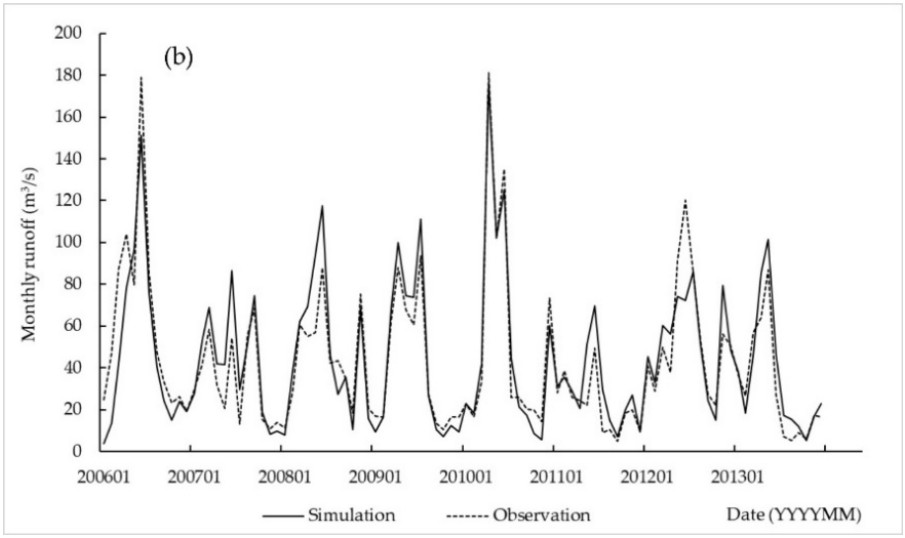

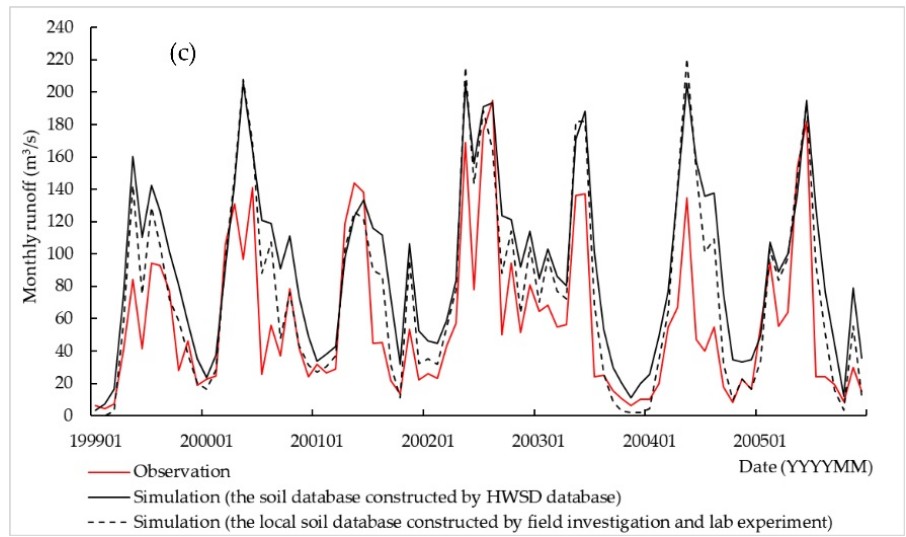

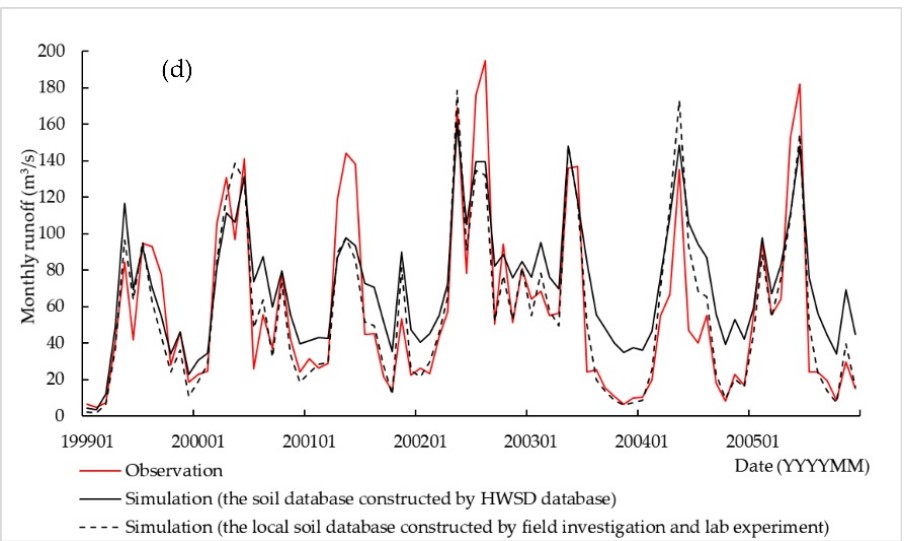

**Figure 3.** Comparison of the observed and simulated monthly runoff of the Zhengshui River Basin. (**a**) The calibration period; (**b**) The validation period; (**c**) The simulated result of two soil databases before calibration; (**d**) The simulated result of two soil databases after calibration.

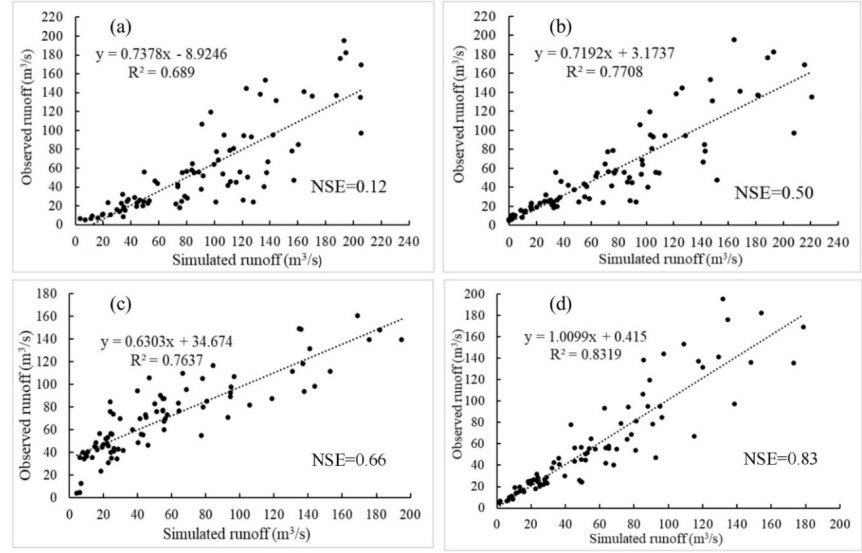

**Figure 4.** Trend lines and scatter diagrams showing monthly observed and simulated runoff of two different databases. Before calibration (**a**) simulated runoff of the HWSD soil database; (**b**) simulated runoff of the local soil database. After calibration, (**c**) simulated runoff of the HWSD soil database; (**d**) simulated runoff of the local soil database.

## 3.2. Effect of Land Use Change on Runoff

Considering that the area of cultivated land, developed land and forestland changed greatly from 1990 to 2010, the changes of these three land use types in each sub-basin and the change of runoff in each sub-basin caused by land use change were analyzed. The sub-basins with the most obvious increases in developed land area are concentrated in the downstream reaches of the river which include the most urbanized areas. The most significant increase of runoff in the basin also occurred downstream, which corresponds to the sub-basins where the area of developed land increased. In sub-basin 41, 44 and 47, the significant conversion of forestland and cultivated land to developed land, resulted in a significant increase in runoff. In sub-basin 49, the change of cultivated land and developed land area was small, so the increase of forestland area was identified as the strongest contributor for the decrease of runoff. In sub-basin 10 and 25, the change of forestland and developed land area was small, the area of cultivated land increased, and the runoff decreased (Figure 5). The increase of forestland area reduces runoff, while the increase of cultivated land and developed land leads to the increase of runoff.

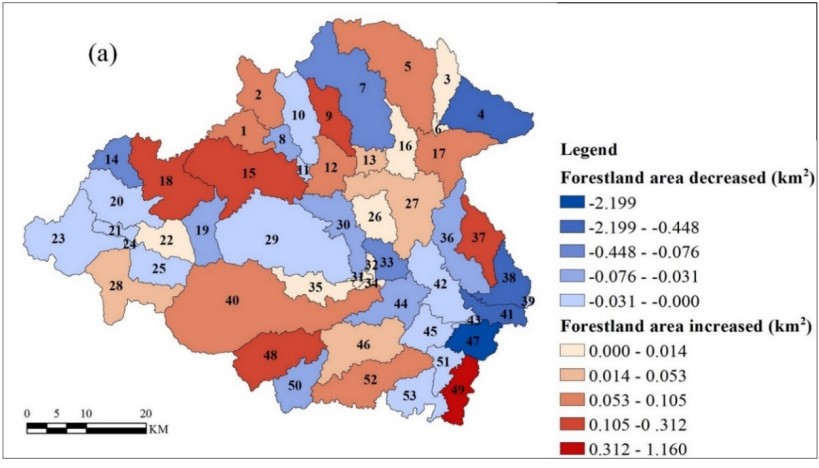

**Figure 5.** *Cont.*

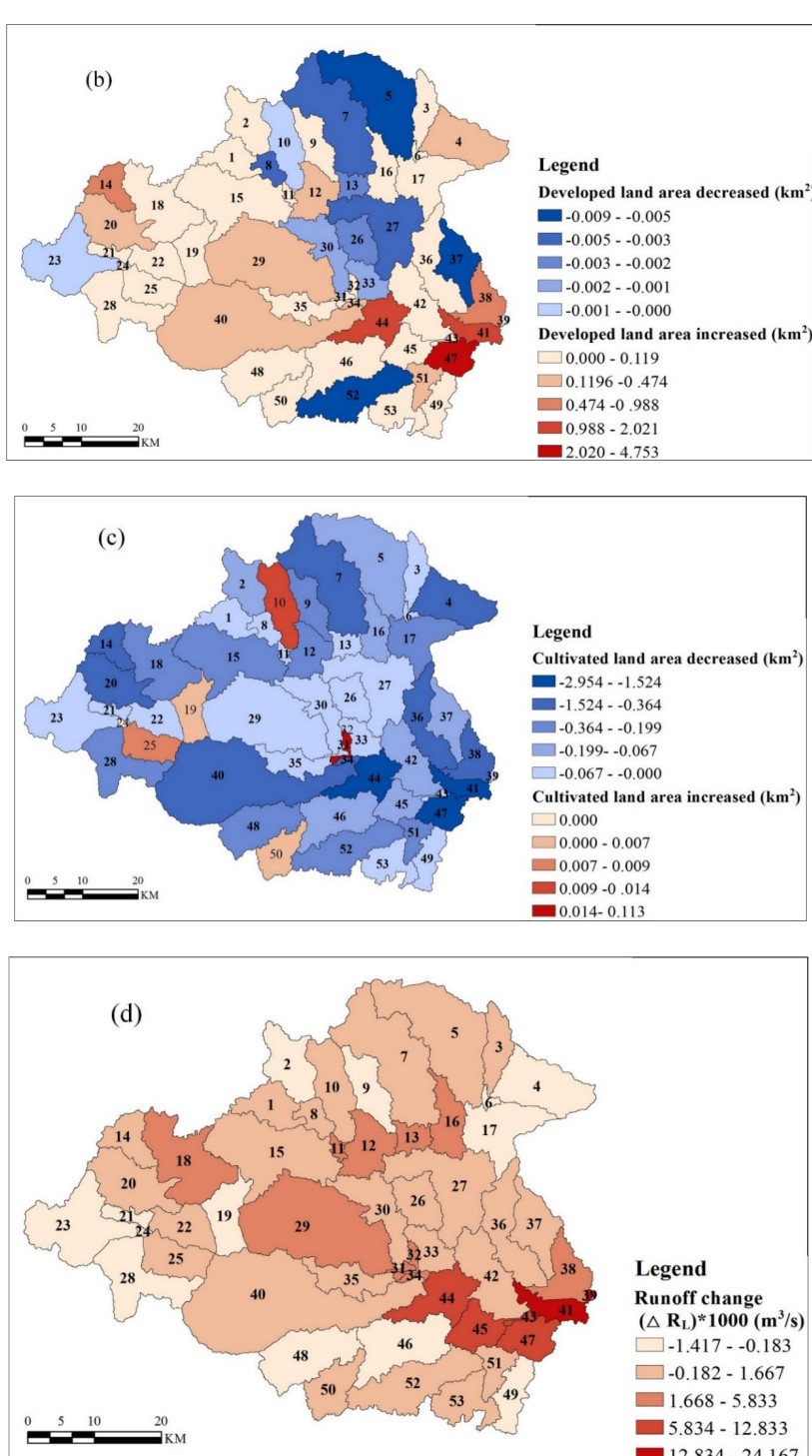

**Figure 5.** Land use and runoff change in each sub-basin from 1990 to 2010. (**a**) The area change in forestland; (**b**) The area change in developed land; (**c**) The area change in cultivated land; (**d**) The simulated average monthly runoff change in each sub-basin under scenarios $S_1$ and $S_3$ which corresponds to the value of $\Delta R_L$ for each sub-basin.

*3.3. Impact of Climate Change on Runoff*

From 1990 to 2010, changes in climate reduced runoff by 60.8 m$^3$/s and climate accounted for up to 100.22% of the change in runoff. Climate change was the main factor affecting simulated runoff change in Zhengshui River Basin. To further analyze the influence of precipitation and temperature on runoff change, change in monthly precipitation, monthly temperature, and monthly runoff in the

basin was calculated from 1990 to 2010. The mean monthly precipitation decreased, the mean monthly temperature increased, and the mean monthly runoff decreased (Table 4). The trend of monthly runoff change in the basin was consistent with the precipitation trend. The trend of temperature change was roughly the opposite of the trend of runoff. Runoff was more sensitive to precipitation changes than temperature changes (Figure 6).

**Table 4.** The change of precipitation and temperature from 1990 to 2010 and monthly runoff change caused by climate change ($\Delta R_C$) from 1990 to 2010.

| Month | Precipitation Change (mm) | Temperature Change (°C) | Runoff Change Caused by Climate Change ($\Delta R_C$) ($m^3/s$) |
|---|---|---|---|
| 1 | −38.0 | 1.7 | −124.9 |
| 2 | −84.6 | 3.3 | −208.5 |
| 3 | −9.1 | −0.1 | −215.8 |
| 4 | 161.0 | −0.8 | 550.2 |
| 5 | 40.0 | 0.6 | 232.8 |
| 6 | 78.8 | −1.9 | 325.5 |
| 7 | −67.2 | 1.2 | −253.3 |
| 8 | 28.8 | 0.7 | −303.4 |
| 9 | −11.4 | 0.1 | 24.7 |
| 10 | −166.7 | −0.3 | −516.4 |
| 11 | −78.1 | −0.9 | −637.3 |
| 12 | 124.4 | 1.0 | 396.6 |
| Average | −1.8 | 0.4 | −60.8 |

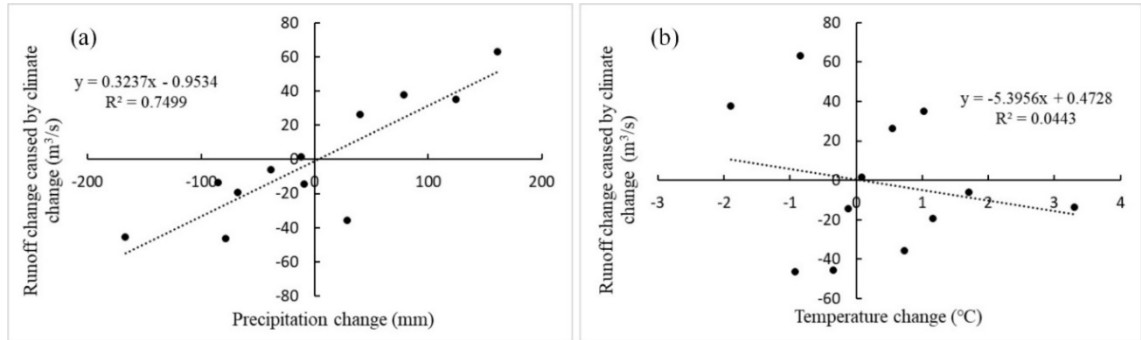

**Figure 6.** Relationship between (**a**) monthly precipitation change and simulated monthly runoff caused by climate change ($\Delta R_C$); (**b**) Monthly temperature change and simulated monthly runoff caused by climate change ($\Delta R_C$).

### 3.4. Runoff Response to Land Use and Climate Changes

The mean monthly runoff simulated in scenario $S_1$ and scenario $S_2$ is 544.9 $m^3/s$ and 484.2 $m^3/s$ respectively, which means that compared with 1990, the mean monthly runoff decreased by 60.7 $m^3/s$ in 2010 under the combined effect of climate change and land use change. Both scenario $S_3$ and scenario $S_1$ model input the same land use data in 1990, and the climate data in 2010 and 1990, respectively. The simulated mean monthly runoff in scenario $S_3$ is 484.1 $m^3/s$, which is 60.8 $m^3/s$ monthly average less than that in scenario $S_1$, caused by climate change from 1990 to 2010. Scenario $S_4$ and scenario $S_1$ model input climate data in 2010, and land use data are 2010 and 1990, respectively. The simulated mean monthly runoff in scenario $S_4$ is 545.0 $m^3/s$, which is 0.1 $m^3/s$ higher than that in scenario $S_1$, caused by land use change from 1990 to 2010 (Table 5). Compared with the change in the mean monthly runoff from 1990 to 2010, the mean monthly runoff decreased by climate change from 1990 to 2010 reached 100.22%, while the mean monthly runoff increased by land use change accounted for only 0.20% (Table 6). Overall, the influence of climate change on runoff was higher than that of land use change in the Zhengshui River Basin during 1990 to 2010. Climate change was the main factor affecting

simulated runoff change in this basin. In addition, land use change can reduce the negative impact of climate change on runoff.

**Table 5.** Simulated monthly runoff of Zhengshui River Basin for four scenarios.

| Month | Simulated Runoff (m³/s) | | | |
|---|---|---|---|---|
| | Scenarios S₁ (Land Use in 1990, Climate in 1990) | Scenarios S₂ (Land Use in 2010, Climate in 2010) | Scenarios S₃ (Land Use in 1990, Climate in 2010) | Scenarios S₄ (Land Use in 2010, Climate in 1990) |
| 1 | 195.7 | 70.8 | 70.8 | 195.9 |
| 2 | 278.6 | 70.1 | 70.1 | 278.7 |
| 3 | 452.2 | 236.7 | 236.5 | 452.4 |
| 4 | 915.1 | 1466.0 | 1465.3 | 915.5 |
| 5 | 777.7 | 1010.6 | 1010.5 | 777.9 |
| 6 | 975.8 | 1301.7 | 1301.3 | 976.1 |
| 7 | 714.3 | 460.9 | 461.0 | 714.3 |
| 8 | 516.1 | 212.6 | 212.6 | 516.2 |
| 9 | 155.9 | 180.7 | 180.6 | 155.8 |
| 10 | 601.3 | 84.9 | 84.8 | 601.4 |
| 11 | 692.0 | 54.7 | 54.7 | 692.1 |
| 12 | 263.9 | 660.6 | 660.5 | 263.9 |
| Mean | 544.9 | 484.2 | 484.1 | 545.0 |

**Table 6.** Impact of land use change and climate change on runoff.

| Scenarios | Mean Monthly Runoff (m³/s) | Runoff Change $\Delta R_{L\&C}$ (m³/s) | Impact of Land Use Change | | Impact of Climate Change | |
|---|---|---|---|---|---|---|
| | | | $\Delta R_L$ (m³/s) | $I_L$ (%) | $\Delta R_C$ (m³/s) | $I_C$ (%) |
| S₁ | 544.9 | | | | | |
| S₂ | 484.2 | −60.7 | | | | |
| S₃ | 484.1 | | | | −60.8 | −100.22 |
| S₄ | 545.0 | | 0.1 | 0.20 | | |

## 4. Discussion

The SWAT model is a physics-based hydrological model and contains parameters of both soil and hydrology. Thus, parameterization of soil characteristics is a key component of the modeling process. Many studies have indicated that the SWAT model is sensitive to soil properties [25,60] and that model estimates can be closer to the actual conditions by providing soil data [61,62]. According to the runoff simulation results of two soil databases, we concluded that both databases provided satisfactory estimates after SWAT model calibration, but the local soil database constructed from field investigation and laboratory experiments produced better estimates. The result is in agreement with previous studies [61,62], which suggested that the soil database constructed from field inventory and laboratory experiments produced simulation results closer to the observed runoff values. Nevertheless, building an accurate local soil database requires time and resources, especially when the study area is a large basin. Therefore, researchers need to be mindful of an application of SWAT model to large basins.

In order to evaluate the relative contribution of climate change and land use change to runoff change, the controlled factor scenarios were used as the SWAT modeling experiments. The scenario analysis involved fixing one factor and changing the other to single out the effect of changing factors on the simulation results [63]. The overwhelming effects of climate change on runoff may be due to the non-linear characteristics of climate change and land use change and SWAT model. Our results also showed that the combined effects were not the summation of the single effects, which also explained why the total contribution rate of climate change and land use change to runoff reduction did not equal 100%. In this study, from 1990 to 2010, climate change has a larger impact than land use change on runoff of the Zhengshui River Basin. Climate change offset the increases caused by land use change and

led to overall runoff decreases. Our research results are different from the results made by Luo et al., that runoff in the Xiangjiang River Basin is more sensitive to land use change [43]. This may be due to the fact that the land use change in the Zhengshui River Basin was relatively moderate from 1990 to 2010.

Considering the spatial heterogeneity within the basin, the relative effects of land use change on runoff in all sub-basins were analyzed. Around urban areas, the expansion of urban areas occupied former forest and cultivated lands and resulted in an increase of the impervious surface area. This reduced infiltration and thus increased runoff in the basin. The result is consistent with the results of Sunde et al. in Hinkson Creek Watershed in the Midwestern U.S [12]. However, Sunde et al. [12] suggested impervious surface change is more important than climate change affecting runoff, which is different from our results. Hinkson Creek Watershed is an urbanized watershed. The area of developed land in the watershed accounts for a large proportion, and urbanization is developing rapidly. Due to the development of urbanization, the impervious area in the watershed has increased by a large extent. From 1990 to 2010, the growth area of developed land was the largest among all land types, but the developed land area accounted for a small proportion in the whole basin, which led to the small contribution to the runoff change. With the development of urbanization and the increase of impervious area, the impact of land use on runoff may exceed that of climate change in the future. The impact of land use change on runoff is variable among sub-basins that cannot be ignored. Hydrological effects should be considered comprehensively in land use planning of the basin.

Previous studies have also shown that climate change reduces runoff [15,64]. This study simply analyzed the impact of precipitation and temperature changes on runoff. From the analysis results, runoff was more sensitive to precipitation changes than temperature changes. Many studies have also reported the same result [65,66]. The expansion of impervious surface in the basin increases runoff by reducing evapotranspiration and infiltration. Climate change affects runoff mainly through influencing precipitation and evapotranspiration. Some research results have shown that evaporation has an important effect on runoff [20], but because there is no measured evapotranspiration data in our study, the simulation effect of evapotranspiration in the study area cannot be evaluated, which is the inadequacy of this study, and it is hoped that the next research work can do further research on this.

## 5. Conclusions

In this study, local soils information based on a field inventory and associated laboratory experiments was used to construct a soils database specifically for application of the SWAT model. Parameter sensitivity analyses and model calibration were conducted using SWAT-CUP. The SWAT model was calibrated and validated using observed monthly runoff data. The simulation results based on a new, detailed soil database constructed by field investigation and laboratory experiments were compared with results based on the relatively coarse HWSD. The calibrated and validated SWAT model was applied to the Zhengshui River Basin to evaluate the impact of changes in land use and climate on runoff under four different scenarios. The local soil database constructed using a field inventory and laboratory experiments performed better in runoff simulations than the soil database constructed from the open HWSD. The combined change of land use and climate indicated a reduction of runoff, which may lead to more severe competition for water resources in the Zhengshui River Basin and a greater impact on the ecological environment of the entire basin. Therefore, rational land use planning and optimizing land use distribution may be effective measures to cope with any negative hydrological effects caused by reduced runoff.

The SWAT model was a useful tool in quantitatively identifying potential impacts of land use and climate changes on hydrological process. The results of this study improved our knowledge and understanding of the runoff response to land use and climate change in the Zhengshui River Basin, and the results serve as a reference for water resources management in the basin.

**Author Contributions:** Conceptualization, L.L. and H.X.; methodology, L.L. and H.X.; software, L.L.; validation, L.L. and H.X.; formal analysis, L.L.; investigation, L.L., H.X., Q.Z., L.D. and G.Z.; resources, H.X. and Q.Z.; data curation, L.D. and G.Z.; writing—original draft preparation, L.L.; writing—review and editing, L.L., H.X. and H.S.H.; visualization, L.L.; supervision, H.X. and Q.Z.; project administration, H.X. and Q.Z.; funding acquisition, H.X. and Q.Z. All authors have read and agreed to the published version of the manuscript.

**Funding:** This research was funded by the Key Project of Education Department of Hunan Province, grant number 19A242; the Basic Work of the Ministry of Science and Technology of China, grant number 2014FY110200; and the Water Conservancy Science and Technology Project of Water Resources Department of Hunan Province, grant number [2017]230-34.

**Conflicts of Interest:** The authors declare no conflict of interest.

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
