# Peer review of "Relative Importance of Land Use and Climate Change on Hydrology in Agricultural Watershed of Southern China"

_sustainability, doi:10.3390/su12166423_

Round 1

Reviewer 1 Report

This is an interesting study, but I am concerned about the way that the authors assume that many of the changes in runoff between 1990 and 2010 are caused by climate change.  There are slight observed differences in temperature and precipitation averages, but the time span is just 20 years and there are just two time data points--1990 and 2010.  Unless the authors can provide a robust set of references justifying such strong conclusions in similar studies, I think they will have to significantly soften their interpretation.  My second general concern is that they provide evidence that a soils dataset based on their own experiments is superior for modeling compared to the FAO dataset but provide no details at all about how their dataset was constructed.  Given the superiority of their dataset, I believe that this is necessary.

Specific comments keyed to lines, tables, and figures are as follows:

L5/6           Numbers here should be superscripts.  Similar errors are also present later in the manuscript, e.g. L26. 

L24                     runoff in this basin

L46                     urban areas are common

L54                     Here and below, when you mention the names of investigators, it is preferred that you insert the citation to the reference directly, and not at the end of the sentence.

L85                     monitoring data from Temple station

L103/104          I am not sure what you mean here—that the Zhengshui River causes the most serious erosion problems of the Xiangjiang River?  Or that the Zhengshui River has some of the most serious problems.  Please clarify.

L105/106          used a mathematical

L108                   model; the

L109                   suitable for application to the

L114                   study of the

L119                   laboratory experiments.  See L313 and throughout.

L144                   The phrase “mainly included” implies that some other data were used.  These should be identified.

Table 1              Please provide more information on the source of land use data.  Where is the data center, and what agency is it affiliated with?

L152/156          I am confused by the use of “red soil,” “waterlogged paddy soil,” and “mountain shrubby-meadow soil.”  Do these apparently imprecise terms correspond to those used by FAO in the HWSD database?  And how were the data from Hunan used to construct your database?  Or did you construct it separately?

L157                   I think you mean USLE_K.  Also, it is probably better to use words to define this factor rather than the abbreviation used in the model.

L158/159          Please provide more information on the scope of this work.  How thorough was your sampling coverage of the basin?  This is quite important, given that this information provided important input information for the model.

L164                   Which were calculated with SPAW?  It is important to be more specific than simply stating “some.”

L170                   Basin based on HWSD.

L179/180          Here and throughout the manuscript, please pay attention to significant figures to the right of the decimal point.  It is sufficient to state 13.0 and 27.9 instead of 13.01 and 27.88.

Table 2              Are you sure about the numbers associated with unutilized land?  It seems unlikely that out of almost 3,500 square km, only 9 ha would be unutilized.  That is a strikingly small number.  Also, I think that this table could be simplified if combined the rows listing square km with those listing percentages.  Thus in the upper left, it would be 1496 square km (43.0%) all on one line, etc.  This would make the table easier to read.

L188                   data were, not data was

L182/196          A 20-year interval is a very short interval of time to assess climate change, which normally occurs over longer intervals.  Can you provide a justification for believing that your analysis of the past 20 years is really measuring climate change?

L220/221          I do not understand what you mean here, especially what you mean by the “actual situation.”  Please clarify.  Also, I would delete the sentence about the 45th sub-basin (readers have no idea where this area is located until they come to Figure 5) and simply refer the reader back to Figure 1, which shows the location.

L285/286          It would be helpful if you would briefly describe what the three sensitive parameters here measure.

Table 4              Delete.  The information in the text is sufficient.

Figure 3            There should be a space between Simulation and the opening parenthesis sign, e.g. Simulation (the and not Simulation(the.

L328/330          I suggest that you refer to the pattern of these three land types rather than the area, which according to Table 2 changed little.  The important finding is not that there is more or less of a given land type in the entire basin, but rather that some sub-basins are seeing increases while others are seeing decreases (shown in Fig. 5).

Figure 5            This figure is carefully drawn, but the numbers identifying the sub-basins are very hard to read. Consider enlarging the entire figure, so that the maps are larger and there is more space for larger numbers.  Also, there appear to be stray lines associated with the small sub-basins near the center of the basin (the red spots in Figure 5c).  This is visible in all of the maps—please check.

L347/348          I do not believe that you have proven cause and effect with respect to climate.  You have shown that there are differences between 1990 and 2010 and that these correlate with differences in weather parameters.  But you have only two data points, and the interval between them is quite short in terms of the interval needed for climate change to reveal itself.  I recommend caution in wording as you consider your findings in Section 3.3.

Table 5              In the absence of statistical analysis of the data on temperature and precipitation in the years preceding 1990, spanning the interval between 1990 and 2010, and those subsequent to 2010, I worry about the interpretation of just two time points with respect to climate.  Also, it should be Precipitation change (mm) and not Precipitation change(mm) in the heading for column 2.

Figure 6            Same concern as above about attributing changes to climate change.

Table 7              I do not understand the derivation of the percentages given here in columns 4 and 5.  I can see how the delta values were calculated in each of these columns but not translate it to percentages.  Are you trying to account for the source of the observed change, i.e. 100% of the delta under scenario 3 can be attributed to climate change?  Or is it something else?  Please revise the narrative to make this absolutely clear.  Also see comment above about spacing out items in parentheses.

L385/387          Are you sure you want to make this statement?  The NSE value for the non-local database  is only 0.12, which is very low.   

L388/389          Need more information about the design of your field inventory and laboratory experiments and that of experiments reported by others.  Are your methods comparable to those given in references 40-41?

L399/400          Why do you expect the numbers, which come from modeling rather than measurements, would exactly equal 100%?

L404/405          Can you be more specific?  How much greater was land use change in the other basin?   

L406/420          I think you are over-interpreting the comparison between your basin and Hinkson Creek.  There are many topographic, hydrological, climate and other differences that could account for the differences.  They are quite understandable.

L409                   consistent with the results of Sunde et al

L410                   in the Midwestern US

L432/451          You can delete this paragraph.  It is just a summary, not a statement of conclusions.

Reviewer 2 Report

Authors evaluate impact of land use and climate change on hydrological processes is with model SWAT for the Zhengshui River Basin.

With current rate of occurring land degradation such studies are relevant and necessary to help prepare for the extreme weather events and consequent natural disasters.

One of the tasks was comparison of the simulation results based on a new, detailed soil database constructed by field investigation and lab experiments (assuming that means laboratory analyses of soil characteristics?) with results based on the relatively coarse (L 119, L 163) (harmonized world soil database) HWSD. This is a very interesting question, one that is rarely addressed.

However, in the methods, authors do not describe what exactly was the improvement of soil data quality – what was the improved grid or number of data points /soil types before vs after.

Furthermore, the first three points in the objectives are addressed in the results, the fourth is included in the first one as a calibration tool. The authors state L321/L322 “Both databases provided satisfactory estimates after SWAT model calibration, but the soil database constructed from field investigation and lab experiments produced slightly better estimates.” Again, what was the improvement in density points?  

Have you evaluated type of urban sprawl (fragmented urbanized areas with existing green areas vs urbanized areas with little or no remaining green/forest patches) that accelerates the runoff? Nature based solutions that serve as detention areas are a measure to fight negative impact of that. How big would the remaining green areas need to be to compensate the negative effect of urban sprawl to make it more sustainable?

Author Response

Response to Reviewer 2 Comments

One of the tasks was comparison of the simulation results based on a new, detailed soil database constructed by field investigation and lab experiments (assuming that means laboratory analyses of soil characteristics?) with results based on the relatively coarse (L119, L163) (harmonized world soil database) HWSD. This is a very interesting question, one that is rarely addressed.

Point 1: However, in the methods, authors do not describe what exactly was the improvement of soil data quality – what was the improved grid or number of data points /soil types before vs after.

Response 1: Thanks for your helpful suggestion. We have added the description of the soil database based on field investigation and laboratory experiments in the revised manuscript (see Lines 147-174). The revised content in our manuscript now reads:

Soil data included a soil type map and some physical-chemical properties of the associated soil. The scale of soil type map used in this study was 1:50000. The classification of soil types of the soil type map is based on Soil Taxonomy in China. There are six levels of Chinese Soil Taxonomy. Considering the unity with the data from soil profile that is necessary parameters for runoff simulation, we adopt the classification of subgroup. The soil in the Zhengshui River Basin was divided into 15 types. The most widespread soil in the Zhengshui River Basin was red soil, with an area of 1,423 km², accounting for 41% of the total area of the basin, followed by waterlogged paddy soil, also accounting for nearly 41% of the total area of the basin. The area of mountain shrubby-meadow soil was the smallest, only 0.06km2, accounting for a small proportion of the total area of the basin. 199 soil profiles were excavated in Hunan Province in order to classify the soil system of Hunan Province, the Hunan Provincial Soil Survey Project. Parameters of soil profile for this study were obtained from the Hunan Provincial Soil Survey Project. Through field observation and analysis, the number of soil layers in each soil type (NLAYERS), the depth from the bottom layer to the surface layer of soil layer (SOL_Z), the maximum root depth of soil profile (SOL_ZMX), and the structure of soil layer (TEXTURE) were recorded. Each soil layer in each soil profiles was sampled and measured in the laboratory to obtain the chemical parameters of soil. The parameters of organic carbon content (SOL_CBN), clay content, silt content, sand content and rock content required by the database were obtained by experiments, especially the particle size was in accordance with the American Standard, which meets the needs of the model, and did not need soil particle size conversion. The USLE soil erodibility factor (K) parameter was calculated by using EPIC model. Groupie of pedohydrology (HYDGRP) was divided according to the size of soil permeability coefficient, and soil permeability coefficient was calculated according to empirical calculation formula of soil permeability coefficient. The parameters related to soil moisture include wet bulk density (SOL_BD), effective water content (SOL_AWC) and Saturated hydraulic conductivity (SOL-K) were calculated with the Soil-Water-Characteristics module (SWCT) in Soil Plant-Atmosphere-Water Model (SPAW). The parameters of anion exchange porosity (ANION_EXCL), maximum compressibility of soil (SOL_CRK), surface reflectance (SOL_ALB) and soil electrical conductivity (SOL_EC) were the default values of the model. With the aforesaid data, the soil database of the Zhengshui River Basin was constructed based.

Point 2: Furthermore, the first three points in the objectives are addressed in the results, the fourth is included in the first one as a calibration tool.

Response 2: In our previous manuscript, we adjusted the structure of the paper, but did not modify the research objectives accordingly. We revised the objectives of this paper in the revised manuscript (see Lines 112-116). The revised content in our manuscript now reads:

Thus, the objectives of this study for of the Zhengshui River Basin were as follows: 1) calibrate the SWAT model for the Zhengshui River Basin; 2) compare the accuracy of the simulation results based on a new, detailed soil database constructed by field investigation and laboratory experiments with results based on the relatively coarse HWSD; 3) quantify the impact of land use and climate changes on runoff by applying the calibrated SWAT model under different land use and climate scenarios.

Point 3: The authors state L321/L322 “Both databases provided satisfactory estimates after SWAT model calibration, but the soil database constructed from field investigation and lab experiments produced slightly better estimates.” Again, what was the improvement in density points?

Response 3: We believe this comment should be the same with the first comment. We have added more details on the construction of soil database based on field investigation and laboratory experiments (see Lines 147-174).

Point 4: Have you evaluated type of urban sprawl (fragmented urbanized areas with existing green areas vs urbanized areas with little or no remaining green/forest patches) that accelerates the runoff? Nature based solutions that serve as detention areas are a measure to fight negative impact of that. How big would the remaining green areas need to be to compensate the negative effect of urban sprawl to make it more sustainable?

Response 4: The question you raised is worth exploring, but it is not considered in our study. It is hoped that the next research work can do further research on this.

Reviewer 3 Report

The article presents good methods and results, and the overall MS quality is good: the structure is well arranged, the presentation is concise and clear .
Overall, I have some minor suggestions to further improve its quality:

  • English language and style are fine but there are some minor spell check in the text that a careful editing could identify and correct;
  • Please standardize the notation for coefficient and the units of measurement (line 21, 26);
  • Please correct USLE K instead of ULSE K (line 157);
  • Please fill in the 'Author Contribution', 'Funding' and 'Conflict of interests' sections, they still contain the standard text.

Author Response

Response to Reviewer 3 Comments

The article presents good methods and results, and the overall MS quality is good: the structure is well arranged, the presentation is concise and clear. Overall, I have some minor suggestions to further improve its quality:

Point 1: English language and style are fine but there is some minor spell check in the text that a careful editing could identify and correct.

Response 1: First of all, thank you for your approval of our article. The entire article has been carefully reviewed again and again, and all spelling and grammatical errors have been revised. And we have checked the language carefully in the revised manuscript. Hope it can meet the need of the journal.

Point 2: Please standardize the notation for coefficient and the units of measurement (line 21, 26).

Response 2: Revised as suggested.

Point 3: Please correct USLE K instead of ULSE K (line 157). 

Response 3: Thanks for your kind reminding. We revised the abbreviations as suggested. It has been revised into "the USLE soil erodibility factor (K)" (see Line 165).

Point 4: Please fill in the 'Author Contribution', 'Funding' and 'Conflict of interests' sections, they still contain the standard text.

Response 4: We have added content about the 'Author Contribution', 'Funding' and 'Conflict of interests' sections (see lines 464-473). The added content in our manuscript now reads:

Author Contributions: Conceptualization, L.L. and H.X.; methodology, L.L. and H.X.; software, L.L.; validation, L.L. and H.X.; formal analysis, L.L.; investigation, L.L., H.X., Q.Z., L.D., and G.Z.; resources, H.X. and Q.Z.; data curation, L.D., and G.Z.; writing—original draft preparation, L.L.; writing—review and editing, L.L., H.X. and H.H.; visualization, L.L.; supervision, H.X. and Q.Z.; project administration, H.X. and Q.Z.; funding acquisition, H.X. and Q.Z. All authors have read and agreed to the published version of the manuscript.

Funding: This research was funded by the Key Project of Education Department of Hunan Province, grant number 19A242; the Basic Work of the Ministry of Science and Technology of China, grant number 2014FY110200; and the Water Conservancy Science and Technology Project of Water Resources Department of Hunan Province, grant number [2017]230-34.

Conflicts of Interest: The authors declare no conflict of interest.

Round 2

Reviewer 1 Report

Thank you for responding to all of the suggestions made in my original review.